# The Role of CT in the Staging and Follow-Up of Testicular Tumors: Baseline, Recurrence and Pitfalls

**DOI:** 10.3390/cancers14163965

**Published:** 2022-08-17

**Authors:** Thibaut Pierre, Fatine Selhane, Elise Zareski, Camilo Garcia, Karim Fizazi, Yohann Loriot, Anna Patrikidou, Natacha Naoun, Alice Bernard-Tessier, Hervé Baumert, Cédric Lebacle, Pierre Blanchard, Laurence Rocher, Corinne Balleyguier

**Affiliations:** 1Department of Radiology, Gustave Roussy, 114 Rue Edouard-Vaillant, 94800 Villejuif, France; 2School of Medicine, University of Paris-Saclay, Cancer Campus, 94800 Villejuif, France; 3Department of Nuclear Medicine, Gustave Roussy, 114 Rue Edouard-Vaillant, 94800 Villejuif, France; 4Department of Oncology, Gustave Roussy, 114 Rue Edouard-Vaillant, 94800 Villejuif, France; 5Department of Urology, Gustave Roussy, 114 Rue Edouard-Vaillant, 94800 Villejuif, France; 6Department of Urology, Kremlin Bicêtre Hospital, APHP, 78 Rue du Général Leclerc, 94270 Le Kremlin Bicêtre, France; 7Department of Radiation Oncology, Gustave Roussy, 114 Rue Edouard-Vaillant, 94800 Villejuif, France; 8Department of Radiology, Antoine-Béclère Hospital, APHP, 157 Rue de la Porte de Trivaux, 92140 Clamart, France

**Keywords:** testicular cancer (TC), germ cells tumors (GCT), seminomatous and nonseminomatous GCT, computed tomography (CT), oncologic diagnostic imaging, retroperitoneal masses

## Abstract

**Simple Summary:**

Testicular cancer (TC) is an uncommon group of tumors affecting predominantly younger males between 15 and 40 years, and accounting for less than 1% of malignancies in men, albeit in the context of an increasing incidence rate over recent decades. Testicular germ cell tumors (TGCT) are the most frequent (90%), and most cases of TGCT are organ-confined at diagnosis. The majority of patients with TGCT have an excellent prognosis, with a 5-year survival rate greater than 95%, and expect to be cured thanks to different risk-adapted treatments such as cisplatin-based chemotherapy, even at advanced stages. It is for this reason that both initial staging and follow-up are essential for appropriate management in initiating adapted therapy as well as treating cases of recurrence, most frequent during the first 5 years.

**Abstract:**

Ultrasound imaging of the testis represents the standard-of-care initial imaging for the diagnosis of TGCT, whereas computed tomography (CT) plays an integral role in the initial accurate disease staging (organ-confined, regional lymph nodes, or sites of distant metastases), in monitoring the response to therapy in patients who initially present with non-confined disease, in planning surgical approaches for residual masses, in conducting follow-up surveillance and in determining the extent of recurrence in patients who relapse after treatment completion. CT imaging has also an important place in diagnosing complications of treatments. The aims of this article are to review these different roles of CT in primary TGCT and focus on different pitfalls that radiologists need to be aware of.

## 1. Introduction

Testicular cancer (TC) represents the most common group of tumors occurring in young adult men aged 18–35 years, and accounts for approximately 1% of malignancies in men [1,2,3]. The main risk factors with established evidence for TC include cryptorchidism, family or personal history of TC [4,5,6]. Organochlorine compounds have also been associated with a risk of developing TC [7].

TC is broadly classified into two major categories: germ cell and stromal cancers. Testicular germ cell tumors (TGCT) are the most frequent (95%) and are subdivided into two histopathological subtypes: approximately 55–60% are seminomatous (SGCT) and 40–45% are non-seminomatous germ cell tumors (NSGCT) including embryonal carcinoma, yolk sac tumor, choriocarcinoma, teratoma and mixed germ cell tumors exhibiting a different degree of embryonic or extraembryonic patterns [8]. In some cases, serum markers can be informative and assist with staging, prognosis, monitoring disease activity and prediction of histopathological type [9].

The majority of patients with TGCT have an excellent prognosis, with a 5-year survival rate greater than 95%, and expect to be cured thanks to different risk-adapted and advanced interdisciplinary treatments such as cisplatin-based chemotherapy, relying on an accurate disease assessment, even at advanced stages [10]. It is for this reason that both initial staging and follow-up are essential for appropriate management in initiating adapted therapy as well as treating cases of recurrence, which are most frequent during the first 5 years.

On one hand, testicular ultrasound using a high-frequency probe (>10 MHz) with Doppler assessment represents the standard-of-care initial imaging [11] to confirm an intratesticular mass prior to orchiectomy, even in the presence of a clinically evident testicular lesion. In addition, it is helpful in assessing patients who present with disseminated disease in whom an occult testicular primary tumor is suspected. It is also necessary to examine the contralateral testicle to exclude other synchronous tumors and identify risk factors for germ cell neoplasia in situ such as small atrophic testis and extensive microlithiasis. Magnetic resonance imaging (MRI) is not recommended in routine use. However, it may be a valuable problem-solving modality for the morphological evaluation and characterization of scrotal masses in patients with unconclusive and equivocal sonographic features, or in case of uncertain location or local invasion [12,13].

On the other hand, computed tomography (CT) plays a crucial role in the initial accurate staging of disease (organ-confined, regional lymph nodes, or sites of distant metastases), in evaluating response to therapy in patients who initially present with non-confined disease, in planning surgical approaches for residual masses, in conducting follow-up surveillance and determining the extent of recurrence in patients who relapse after treatment completion [13,14,15]. CT imaging also has an important place in diagnosing complications of chemotherapy or surgery treatments.

The aims of this article are to review these different roles of CT in primary TGCT and focus on the different pitfalls that radiologists need to know.

## 2. Baseline and Staging

The suspected diagnosis of TC is often identified by physical examination and testicular ultrasound to confirm the intratesticular location of the mass. However, the diagnostic documentation of TC is based on histopathology after orchiectomy. In case of a suspected TC, orchiectomy including division of the spermatic cord at the internal inguinal ring is the standard-of-care management and is therefore both a diagnostic and therapeutic procedure [12]. A scrotal approach ought to be avoided because of a higher local recurrence rate. In patients with advanced metastatic TGCT and/or those with impeding organ failure, orchiectomy can be postponed after completion of the chemotherapy or between treatment cycles. Biopsies of the contralateral testis should be discussed for high-risk patients for a second TGCT, i.e., those aged < 40 years with a small atrophic testis and/or extensive microlithiasis [12].

Upon a diagnosis of a TGCT, baseline assessment of disease extent is crucial prior to initiating therapy. Several classifications are used in the literature for the initial assessment of TGCT. They are widely staged by the TNMS (Tumor-Nodal-Metastasis-Serum Tumor Marker) system set forth by the American Joint Commission on cancer (AJCC) staging [16] (Table 1), combining the local tumor extent on surgical pathology after orchiectomy, regional and distal lymph nodes and distant metastases involvement, and finally, post-orchiectomy serum marker levels, including alpha-fetoprotein (AFP), Human Chorionic Gonadotropin (HCG) and Lactate Dehydrogenase (LDH).

While the T stage is determined histopathologically [17], CT imaging plays a major role in determining the N node extent (retroperitoneal nodes) and metastatic M disease (non-retroperitoneal nodes such as supra-clavicular, mediastinal nodes; and non-nodal distant metastases) and is recommended in all patients with a diagnosis of TGCT. Combining these TNM-S components, the AJCC classification (Table 2) categorizes TGCT into three major groups: tumor limited to the testis (stage I), retroperitoneal lymph nodes extent (stage II) and distant disease (stage III). Thus, categorization into these three stages is impossible without contrast-enhanced CT imaging of the thorax, abdomen and pelvis, which is the reference standard for the evaluation of nodes and metastases assessment.

The International Germ Cell Tumor Consensus Collaborative Group (IGCCCG) classification is commonly used for patients with advanced TGCT and stratifies patients into good, intermediate and poor prognostic groups [18] (Table 3). This latter classification is based on different criteria: primary TC histology, location of primary tumor, presence or absence of non-pulmonary metastases, and levels of post-orchiectomy serum markers. It slightly differs between SGCT and NSGCT. This classification was recently updated with contemporary outcomes and refined risk stratification for metastatic TGCT [19,20].

TGCT spread occurs via lymphatic (regional disease for retroperitoneal lymph nodes, and metastatic disease including other lymph node areas such as thoracic, supra clavicular and pelvis regions) and vascular (metastatic disease) patterns, which defines N and M stages.

### 2.1. Retroperitoneal Lymph Nodes: N Stage

The retroperitoneal lymph nodes are the most common sites for TGCT (Figure 1). CT imaging remains the standard modality used for assessment of lymph node involvement, distinguishing between stages II (retroperitoneal nodes considered as regional lymph nodes) and III (supra-diaphragmatic nodes considered as metastatic disease) [21].

TGCT most commonly spread to retroperitoneal lymph nodes, especially in periaortic lymph nodes because of their lymphatic and venous pathway dissemination [21,22,23,24,25]. Thanks to the knowledge of the patterns of lymphatic and venous drainage pathways, the sites of lymph nodes can be explained, and this predictable fashion throughout the lymphatic system may improve the accuracy of detection and analysis of lymph nodes by radiologists. The spread follows the lymphatic channels and the gonadal veins along the spermatic cord, through the inguinal ring to enter into the retroperitoneum, unless the lymphatic drainage has been altered by prior procedures such as inguinal or scrotal surgeries, or cryptorchidism, tunica vaginalis invasion, or previous retroperitoneal lymph node dissection. The right gonadal vein drains into the inferior vena cava, so right-sided TGCT usually involve aortocaval or paracaval nodes, whereas the left gonadal vein drains into the left renal vein, so left-sided TGCT typically metastasize to left periaortic nodes, most often below the left renal hilum. In case of the absence of ipsilateral lymphadenopathy, contralateral spread is unusual, and histological analysis is recommended prior to initiating therapy [26]. Moreover, pelvic lymphadenopathy is unusual in the absence of periaortic lymphadenopathy, unless the lymphatic drainage has been previously altered.

It is important to note that any nodal disease superior to the level of the renal hila is considered as metastatic disease and should be included in the M staging.

CT demonstrates an excellent sensitivity to identify lymph nodes, thanks to its excellent spatial resolution; however, it cannot formally differentiate benign from infiltrated lymph nodes, especially for smaller nodes [21]. The lymph node short diameter is therefore the most standard measurement considered to distinguish benign from metastatic nodes. The short axis size is used to discriminate benign and malignant lymph nodes (N0 versus N1 disease stages), whereas the greatest dimension is the reference to determine N staging. However, the size cut-off is not formally established without controversy and remains a challenge, influencing the sensitivity and specificity of CT [27]. Indeed, different studies [27,28,29,30] have been performed to assess the most appropriate short axis lymph node size cut-off, as an indicator of malignant involvement, in comparison with the reference standard which is the lymphadenectomy (retroperitoneal lymph node dissection). With a threshold of 4 mm or greater to consider positive lymph nodes, the sensitivity is excellent (>90%); however, the specificity is lower, approaching 58%. Conversely, with a larger cut-off size (10 mm), the specificity approaches 100%, but the sensitivity becomes very limited, between 37 and 47% according different studies [27,30]. A threshold decreased to 7–8 mm or greater would provide a specificity and sensitivity of 70%. For each cut-off, there is a substantial overlap between benign and malignant lymph nodes. However, distinguishing benign from malignant lymph nodes has a critical clinical aspect as it determines disease stage and is decisive in defining further treatment.

Because of the lack of a validated consensus, the short axis diameter usually recommended to consider a lymph node suspicious is 8 mm, especially in higher risk patients with lymphovascular invasion at histopathology of orchiectomy specimen or a high proportion of embryonal subtype. Laterality of the primary tumor is also an additional clue. With this CT size cut-off for retroperitoneal nodes, the AUC is the most significant with a sensitivity and specificity approaching 70% [30], although 30% have micrometastatic lymph nodes.

TGCT have a high micrometastatic lymph node incidence; therefore, even if the size is below the cut-off, radiologists need to analyze morphologic features such as the shape (round or spiculated) and the enhancement (heterogeneous, central necrosis, etc.), which suggest positive lymph nodes [31]. In case of stage I disease but with lymph nodes less than 7–8 mm, repetition of the CT over 4–6 weeks can be informative as to inflammatory, normal or micrometastatic lymph nodes, thanks to the CT evolution.

The use of MR imaging [32] has an equivalent performance to that of CT imaging for diagnosis of lymph nodes, and it can be an important alternative because it avoids exposition of patients to ionizing radiation in patients with TGCT who are young men with a high likelihood of cure, but there is a relative lack of universal availability of this modality and greater costs. MRI is evidently the modality of choice in case of CT contraindications such as iodine allergy or severe renal failure.

### 2.2. Metastatic Disease: M Stage

CT of thorax, abdomen and pelvis remains the standard modality currently performed to assess metastatic disease.

The M stage defines the presence of distant metastatic disease, including distant lymph nodes outside the retroperitoneum. TGCT can spread above the diaphragm to the posterior mediastinum via the thoracic duct, it is therefore very important to analyze this region (Figure 1). NSGCT have a more random spread involving the anterior mediastinum, aortopulmonary window, hilar, supraclavicular and neck lymph nodes [33].

This M stage is most often present in choriocarcinoma and yolk sac tumors because of their hematogenous dissemination. The most common metastatic site of solid organs is the lungs. Most of the time, lung metastases appear as multiple lung nodules (Figure 1).

Other sites include liver, bones and rarely, testicular metastasis may occur to adrenals, kidneys, spleen, pleura, pericardium, peritoneum or retroperitoneum in very advanced disease (Figure 2). In case of large retroperitoneal lymph nodes, an extension may occur into the inferior vena cava in the form of tumor thrombus (Figure 2).

Brain imaging is not routinely performed but is indicated in symptomatic patients, in patients with high risk factors such as choriocarcinoma type tumor (likely hemorrhagic metastases), serum marker HCG > 10,000, or in advanced metastatic disease, including supradiaphragmatic nodal disease (Figure 3).

## 3. Follow-Up and Recurrence

### 3.1. Imaging for Active Surveillance

Approximately 80% of patients with SGCT present with stage I disease, with a survival rate of 99%. Active surveillance is the most common management pathway followed after orchiectomy. Adjuvant chemotherapy with one course of carboplatin AUC7 ought to be discussed with patients unwilling or unable to undergo surveillance or higher-risk patients defined by the presence of one or both risk factors (size of the tumor greater than 4 cm or rete testis invasion). In low-risk stage I NSGCT (absence of lymphovascular involvement), active surveillance is also the preferred management attitude. Adjuvant chemotherapy with one course of BEP (Bleomycin, Etoposide, Cisplatin) is an alternative. In cases of stage I NSGCT with high risk, adjuvant chemotherapy ought to be discussed but surveillance can be an alternative [34,35].

Many variables must be considered by the oncologist before opting for active surveillance. The most important point is the compliance and lucidity of the patient because of the frequency and rigor of imaging and examinations. Surveillance monitoring may differ slightly from institution to institution because of the absence of single generally accepted consensus, but it begins with more frequent imaging in the first 3 years and slowly decreases thereafter, as the annual relapse rate drops. At our institution, the protocol includes physical examination and serum marker levels assessment (AFP, HCG, LDH) every 4 months for the first 2 years, every 6 months for year 3 and then annually, and CT of the thorax, abdomen and the pelvis every 4 months within the first two years, every 6 months for year 3 and then annually through to year 5. The impetus for this strategy is that in patients with SGCT, depending on the presence or absence of risk factors, approximately 12–20% of individuals who undergo active surveillance will relapse within 5 years [36]. NSGCT has a cumulative relapse rate of 30%, ranging from 14% (low risk) to 40–50% (high risk) depending on the risk of lymphovascular involvement [37]. An additional argument for active surveillance is that cure rates after relapse are excellent, estimated to be 98% for NSGCT and 100% for SGCT thanks to very effective salvage chemotherapy regimens [38]. Recent studies demonstrate that surveillance with CT imaging provides the same overall survival rates for stage I disease without exposing patients to secondary effects of chemotherapy, radiation therapy or lymph node dissection.

CT plays a pivotal role in active surveillance and in identifying recurrence; the most common site of relapse is the retroperitoneum. It is very important to detect early recurrence because 30% of stage I TGCT are staged incorrectly on initial CT assessment, owing to lymph nodes micrometastases.

### 3.2. Response to Therapy and Post Therapeutic Changes

Patients with stage II or III TGCT are commonly treated with poly-chemotherapy regimens. CT imaging remains the standard modality for monitoring therapeutic response by measuring the diameter of the different lesions over several consecutive examinations [14,15]. The EGCCCG guidelines recommend that CT assessment must be performed at the end of first-line chemotherapy. An intermediate assessment must also be performed earlier in cases of poor-risk NSGCT patients treated according the GETUG 13 protocol [39]. For metastatic SGCT treated according to the SEMITEP protocol, a PET-CT is also performed after two cycles to guide de-escalation [40]. To reduce radiation exposure, a contrast-enhanced PET-CT is preferrable to avoid performing PET and contrast-enhanced CT almost at the same time, except for cases when detailed, fine-section analysis of the lung parenchyma is needed. In case of contraindication for injection of an iodinated contrast agent, an abdominopelvic MRI with a gadolinium injection can be performed combined with a chest CT scan without injection.

For metastatic disease, the standardized Response Evaluation Criteria in Solid Tumors (RECIST 1.1) is the reference quantitative reporting method in TGCT, as in most other solid tumors, which can determine a stable disease, an unequivocal progression, or a complete or partial response to therapy [41]. On the baseline CT, a maximum of five target lesions are selected and measured by the radiologist (maximum of two per organ) on axial CT images. Visceral targets must have a long axis >10 mm, target lymph nodes must have a short axis >15 mm, and bone lesions must have a soft tissue component > 10 mm. The radiologists should avoid ill-defined lesions, necrotic lesions and confluent lesions to provide reliable measurability on follow-up exams. All other lesions that cannot be quantitatively measured are considered as non-target lesions and reflect the overall tumor burden.

Reduction in the size of lymph nodes and metastases indicates a positive response to therapy even if residual masses persist. A more robust assessment of therapeutic response is provided by decrease in serum marker levels when these are informative.

The features of residual masses can also help assess response to treatment. For example, cystic and fatty changes seen on CT usually correspond to mature differentiated teratoma [42].

However, after treatment, it is not possible for CT imaging to distinguish benign lesions such as fibrotic or necrosis residual lesions from viable tumor [43].

In cases of NSGCT, a complete response is defined as residual masses less than 1 cm. Surgical resection is indicated in residual masses greater than 1 cm. Studies have shown that such residual masses have mature teratoma components in up to 40% of patients and residual viable disease in up to 20%, and only fibrosis remaining in 40%. Teratomas need to be removed because they are not responsive to chemotherapy and may undergo a malignant transformation.

In cases of SGCT, residual masses less than 3 cm are likely to have sustained complete fibrosis and necrosis. Indeed, seminoma are very sensitive to radiotherapy and chemotherapy, and surveillance is recommended. An FDG-PET CT is recommended in residual masses >3 cm to select patients according to the presence or not of metabolic activity providing evidence of active tumor tissue and, as such, to identify patients who may benefit from surgical resection and those who can continue surveillance [44].

Thanks to its high spatial resolution and its anatomical precision, CT imaging is essential in planning the operative approach with the surgeon, establishing a cartography of the different sites of residual masses and specifying anatomy variants such as retroaortic left renal vein for retroperitoneal lymph node dissection.

In patients with complete radiological response after chemotherapy, the end-of-treatment CT is considered as the nadir exam for the follow-up. After surgery of residual masses, it is also fundamental to perform a post-operative CT study as baseline, especially in differentiating lymphocele versus teratoma which can present similar features in CT examination.

### 3.3. Recurrence

As with any type of cancer, CT examination is essential to detect early recurrence [14]. Relapse rates vary according to histology, disease stage and treatment modality.

In patients with stage I disease, overall, approximately 30% of patients will relapse in case of active surveillance [45]. The relapse rate is highest within the first three years (80% of relapses occur during the first year, 90% within 2 years) and decreases annually until it becomes lower than 1% after year 5. Late recurrence over 10 years is very rare but may also occur [46]. The most common sites of relapse are the retroperitoneum in approximately 60%, and the lungs in 25% [46].

In cases of active surveillance following orchiectomy, relapse rates are 15–20% in stage I seminoma and 20–50% in non-seminoma stage I [47].

As mentioned above, in stage I SGCT, there are two major factors considered as recurrence risk factors: size of the tumor greater than 4 cm and infiltration of the rete testis. If neither risk factor is present, the risk relapse is up to 8%; if one factor is present, it increases to 15–20%, and approximately 20% if both are present. In stage I NSGCT, the presence of vascular invasion of blood or lymphatic vessels is considered as the most important predictor of micrometastases and subsequent recurrence. If this risk factor is present, it increases the relapse rate from 20% to 50% [48].

In patients with seminoma stage I disease after single-agent carboplatin, five-year relapse rates is <6%, with more than 80% of relapses occurring in the abdomen [49].

In patients with non-seminoma stage I disease after adjuvant chemotherapy, the rate of relapse is <5% compared to 50% if possessing high-risk factors.

In case of disseminated disease (seminoma or non-seminoma), once curative treatment has been achieved, relapse rates of up to 10% might be expected [50].

### 3.4. Surveillance Imaging Modality: CT versus MRI

Because of repeated CT surveillance in young and otherwise healthy TGCT patients, there are concerns about the cumulative radiation exposure and secondary malignancies [51]. In attempts to reduce this exposure, international guidelines list MRI of the abdomen and the pelvis as an alternative to abdominal CT. MRI benefits from the lack of iodine injection (in case of CT contraindications such as iodine allergy or severe renal failure) as well as of ionizing radiation. Different studies have confirmed comparable results between CT and MRI in detection of retroperitoneal lymph nodes and other infra-diaphragmatic metastases [32,52,53,54], especially in experienced centers [53]. The radiation dose from a CT depends on several parameters such as the scanner model, scan protocol and patient size [55]. Despite the increased use of CT dose-saving protocols and limitations on field of view to reduce the radiation exposure, the risk could be eliminated by the use of MRI.

However, despite its encouraging results, MRI has not yet gained the status of a standard procedure because it is more time consuming, of higher cost, and often less available than CT, limiting its role to a problem-solving tool for selected cases [56]. Moreover, lung parenchyma cannot be assessed by MRI. Low-dose chest CT displays chest pathology better than X-ray.

In the future, follow-up may be performed by low-dose chest CT combined with abdomino-pelvic MRI in early stage testicular cancer. However, MRI follow-up needs to be validated from further studies to be a standard-of-care procedure in guidelines, especially if MRI becomes more available, if costs decrease and if insurance companies would undertake the financial burden of this change.

### 3.5. Surveillance Programs and Future Perspectives

According the EAU guidelines [12], for seminoma stage I on active surveillance or after adjuvant therapy (carboplatin or radiotherapy), abdomino-pelvic CT is recommended every 6 months within the first two years, then once at 36 months and at 60 months. Chest imaging is performed only when there are symptoms. For non-seminoma stage I on active surveillance, abdominopelvic CT is recommended at 6, 12, 36 and 60 months. Chest radiography is also recommended every 6 months within the first two years, but low-dose chest CT is rather used. For advanced disease with complete remission, CT of the chest, abdomen and the pelvis is recommended every 6 months within the first year, and then at 24 months, 36 months and 60 months.

However, surveillance schedules differ throughout the word with respect to number and frequency of imaging as well as use of thoracic CT versus X-ray [12]. Depending on the guidelines used (NCCN or ESMO), 5–17 CT exams are recommended during the course of follow-up.

The concern about more frequent CT imaging in earlier diagnosis of relapse and whether this has any outcome on survival rate remain controversial and must be weighed against financial costs and radiation exposure. By using MRI instead of CT and reducing the number of scans, in patients with low-risk of recurrence, early relapses are still detected, and these approaches reduced radiation exposure. For example, the TRISST (Trial of Imaging and Schedule in seminoma Testis) trial compared different monitoring approaches in early stage seminoma patients [57]. These patients were monitored with either the standard follow-up of 7 CT scans, 3 CT scans, or the same regimes with MRI scans. This trial has shown that the outcomes of patients receiving MRI and lower frequency of imaging (at 6, 18 and 36 months) were not worse than those of other patients and that scanning patients beyond year 3 may not be necessary. However, it is not applicable to men with a higher risk for relapse. Moreover, the compliance of the patient is primordial, especially if the monitoring is less frequent, because some patients might lose touch with the doctors.

In the future, most guidelines, especially in poor-risk for relapse patients, are likely to reduce the frequency of imaging during follow-up and replace CT with MRI of the abdomen and the pelvis to minimize irradiation, but further studies are needed to validate these results.

## 4. Pitfalls

### 4.1. Small-Volume Disease

#### 4.1.1. Supraclavicular Lymph Nodes

It is very important to examine the left supraclavicular region, because it is a classic localization of lymph nodes and is often missed because of artifact shoulders (Figure 4).

#### 4.1.2. Small Retroperitoneal Lymph Nodes

As mentioned above, in cases of stage I disease with lymph nodes of 7–8 mm, repetition of the CT in 4–6 weeks can provide information about inflammatory, normal or micrometastatic lymph nodes thanks to the CT evolution.

##### Positive Lymph Nodes

TGCT have a high spread to micrometastatic lymph nodes; therefore, even if the size is under the cut-off, radiologists need to analyze morphologic features such as the shape (round or spiculated) and the enhancement (heterogeneous, central necrosis), which suggest positive lymph nodes (Figure 4).

##### Negative Lymph Nodes

In some cases, lymph nodes can be inflammatory because of gastro intestinal conditions such as mesenteric lymph nodes.

#### 4.1.3. Different Features of Lung Metastases

Most of the time, they appear as multiple small nodules in TGCT. However, lung metastases can appear as different features such as pulmonary miliary disease, small excavated nodules or irregular nodules (Figure 4), and it can be complicated to distinguish metastases from infection or granulomatosis disease. Perifissural lung micronodules are most of time considered as intrapulmonary lymph nodes, but in advanced disease, they can be metastatic disease with lymphatic spread.

### 4.2. Post-Chemotherapy Lymph Nodes Changes

After chemotherapy, lymph nodes reduce in size (Figure 5) and may show some changes such as necrotic features with possible small air bubbles. This is a classical aspect and needs not be considered secondary to a fistula of the digestive tract.

As mentioned above, no imaging can reliably differentiate viable tumor, necrosis or fibrosis. Approximately, 40% of patients have residual mature teratomatous masses after induction chemotherapy, 40% of masses consisting of fibrosis/necrosis and 20% consisting of viable malignancy.

It is classical to see calcifications in residual masses, which is most of the time a sign of good response.

An FDG-PET CT is performed after chemotherapy in case of seminoma to assess disease response (Figure 5), but it is not recommended for NSGCT because FDG uptake of mature teratoma is similar to that of necrotic and cystic masses.

### 4.3. Teratoma, Growing Teratoma, and Transformed Teratoma

#### 4.3.1. Mature Teratoma

Mature teratoma masses feature cystic or necrotic contents and are often multiloculated with enhancing septations. Fat and calcifications may be present (Figure 6).

#### 4.3.2. Growing Teratoma

Growing teratoma is a rare entity (incidence ranging from 1.9 to 7.6%) first described by Logothetis [58], affecting patients with NSGCT and characterized by recurrent enlargement of metastatic masses despite appropriate systemic chemotherapy, during or after chemotherapy (even many years after tumor onset) despite normalized or decreasing serum markers. Histological analysis is the only way to confirm the diagnosis, revealing the presence of mature teratomatous elements without other components of viable tumors. Surgery remains the only curative treatment [59,60,61].

Two leading theories have been proposed: first, chemotherapy destroys only the immature malignant cells, leaving the teratoma cells intact; and second, chemotherapy kinetically promotes the de-differentiation of malignant cells toward benign mature teratoma.

A growing teratoma syndrome must be suspected in patients with the three following criteria (Logothetis’ criteria): increasing size of masses during or after systemic chemotherapy; normalization of previously elevated serum tumor markers; mature teratoma in pre-existing histological analysis.

Chemotherapy and radiotherapy are known to be ineffective treatments for mature teratoma. Therefore, even though lesions are histologically benign, adequate and total radical surgery is the only curative treatment, and it must be complete since it determines the prognosis. Growing teratoma recurrence is reported in 72–83% of patients with partial resection and in 4–17% despite a complete resection. Mature teratoma if not treated appropriately may contribute to relevant symptoms and morbidity because of massive local tumor growth compressing vascular and visceral surrounding structures as well as a malignant transformation [59,60,61].

Some CT scan features are commonly associated with the presence of growing teratoma: better circumscribed margins, expanding cystic and necrotic appearance of the lesions in the pretherapeutic imaging, as well as low densities on post therapeutic imaging, punctate or curvilinear calcifications (Figure 6).

Growing teratoma is most often observed in the retroperitoneum, but other organs can be involved such as lungs, supra clavicular lymph nodes, inguinal nodes, mesentery or liver.

#### 4.3.3. Transformed Teratoma

Malignant transformation of teratoma is defined as the transformation of a somatic teratomatous component of a GCT to a non-germ cell tumor malignant phenotype, with the most frequent transformed histologic types consisting of sarcoma (most frequently rhabdomyosarcoma), adenocarcinoma and primitive neuroectodermal tumors [62]. It is a very rare entity occurring in approximately 3 to 6% of metastatic GCT that commonly arises from mediastinal NSGCT rather than gonadal primary tumors [63]. The radiological signs are non-specific. The prognosis is worse than that of patients without malignant transformation. It is a highly aggressive tumor resistant to platinum-based chemotherapy. Surgical treatment remains the standard of treatment for transformed teratoma and some chemotherapy regimens may be effective if directed towards the transformed histology [64].

### 4.4. False-Positive Diagnoses

#### 4.4.1. Mediastinal Metastatic Lymph Node or Differential Diagnoses?

Mediastinal and hilar lymph nodes are not always metastatic disease and can correspond to granulomatous disease such as sarcoidosis or tuberculosis. Thorax CT can provide clues to distinguish them via the analysis of the pulmonary parenchyma. Other differential diagnoses are possible such as bronchogenic cysts (Figure 7).

#### 4.4.2. Retroperitoneal Lymph Node or Lymphocele?

Lymphocele is a cyst filled by lymph fluid that can occur after retroperitoneal lymphadenectomy. Most of the time, lymphoceles decrease in size over months after surgery and can resolve completely within a few months.

#### 4.4.3. Teratoma versus Lymphocele

These entities can present similar features in CT examination. Lymphocele is present after surgery and will decrease in size with a cystic component most of the time (Figure 7), but it can have inflammatory reactions with a greater volume and a thicker wall. However, teratomas are not present on the immediate post-operative CT in cases of complete resection, appear during follow-up examinations and tend to increase in size. They are also cystic, but most of the time, there is tissular component. Sometimes, it is very difficult to differentiate lymphocele from teratoma, especially without prior CT to assess evolutions. MRI and contrast injection US can help to discriminate these two entities.

### 4.5. Post-Treatment Complications

#### 4.5.1. Bleomycin-Induced Pneumonitis

In patients treated with Bleomycin, etoposide and cisplatin (BEP) chemotherapy, bleomycin-induced pulmonary toxicity is a well-known side effect, observed in approximately 10% of patients treated (Figure 8). It can occur during bleomycin treatment leading to treatment discontinuation, or can develop after a free-treatment interval of weeks to months. It can be fatal in 1–3% of affected patients. Risk factors are the cumulative dose of bleomycin, smoking, impaired renal function, older age. The radiological signs are non-specific and consist of a newly developing interstitial pneumonitis since the start of the bleomycin treatment, not explained by infection or previous fibrosis. It can be unifocal, multifocal, uni- or bilateral and can affect several lobes. Most of these radiological changes appear to resolve on follow-up CT scans [65].

#### 4.5.2. Post-Surgical Complications

Surgical complications are an ongoing issue in post-chemotherapy retroperitoneal lymph node dissection for surgical resection of residual masses [66]. This surgery is technically much more challenging than primary resections because of desmoplastic reactions as an effect of chemotherapy.

The most frequent complications are lymphatic complications such as lymphatic leakage due to the injury of lymphatic channels during retroperitoneal lymph node dissection, most commonly lymphocele and chylous ascites [67]. Chylous ascites is the accumulation of lymph fluid in the peritoneal cavity (Figure 8). Lymphocele is a cyst filled by lymph fluid, and inflammatory reactions are possible, which can account for a greater size during follow-up. Usually asymptomatic, lymphoceles invariably decrease in size over time after surgery and can resolve completely within a few months without any treatment. A surgical intervention may be required if the lymphocele is symptomatic, infected, or compresses vital structures. Body mass index and the number of resected lymph nodes are well-known factors impacting the incidence of lymphocele.

#### 4.5.3. Residual Disease after Surgery

Post-operative CT exam is equally to identify residual disease after surgery. Resection can be incomplete sometimes, especially in difficult areas (renal hilum).

### 4.6. Retroperitoneal Differential Diagnoses

A large number of neoplastic entities may present as retroperitoneal lymph nodes. In a young male, four major diagnoses must be mentioned: TGCT, lymphoma, ganglioneuroma and sarcoma [68,69].

Lymphoma typically presents with retroperitoneal non-compressive lymph nodes which tend to form confluent soft tissue masses surrounding the vessels, without compression or occlusion, with a mild uniform homogeneous enhancement.

Ganglioneuroma is a rare benign tumor arising from the neural crest cells from the sympathetic nervous system, typically seen as a well-circumscribed retroperitoneal mass, surrounding vessels without compression or occlusion, with a low density (approximately 30HU), weak enhancement, and may contain some punctate calcifications [70].

Sarcoma often presents as a large mass with presence of pseudocapsule, which represents compressed tissue of normal adjacent organs. Liposarcoma is the most common retroperitoneal sarcoma characterized by macroscopic fat component with smooth margins.

Of course, non-neoplastic diagnoses must be mentioned such as tuberculosis, Erdheim–Chester disease, retroperitoneal fibrosis, extramedullary hematopoiesis, and cystic lesions such as lymphangioma or lymphocele, amongst others.

## 5. Conclusions

While rare, TC typically affects young men and has an excellent prognosis for most patients due to different risk-adapted and advanced multidisciplinary treatments, which rely on an accurate disease assessment, even at advanced stages.

Diagnostic imaging with CT therefore plays a crucial role in the initial accurate assessment of N and M stages, which is relevant to prognosis. CT also plays a fundamental role in evaluating response to therapy with typical post-therapeutic changes, in planning surgical approaches for residual masses, in conducting follow-up surveillance and recurrence, in determining the extent of recurrence in patients who relapse after treatment completion, and in diagnosing complications of treatments.

Many published follow-up recommendations might expose young TGCT patients to unnecessary radiation, which is of concern with its inherent risk of secondary malignancy. In the future, most guidelines, especially in poor-risk for relapse patients, are likely to reduce the frequency of CT scans during follow-up and replace CT with MRI of the abdomen and the pelvis to minimize irradiation, but there is nowadays a relative lack of universal availability of this modality and greater costs.

## Figures and Tables

**Figure 1 cancers-14-03965-f001:**
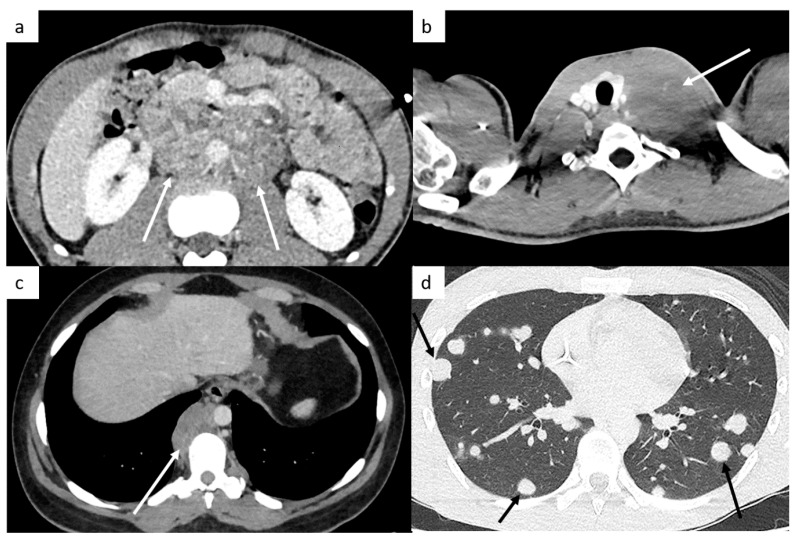
Most common sites of lymph nodes and metastases in testicular cancer. Abdominal computed tomography (CT) shows retroperitoneal paraaortic lymph nodes (**a**, arrow). Thoracic CT reveals left supraclavicular lymph node (**b**, arrow). Thoracic CT demonstrates posterior mediastinal lymph nodes (**c**). Thoracic CT scan shows lung metastases appearing as multiple lung nodules (**d**).

**Figure 2 cancers-14-03965-f002:**
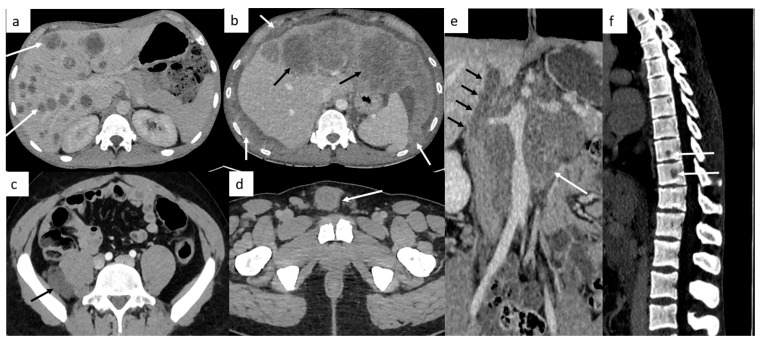
Other sites of metastatic disease. In advanced stages, abdominal CT can show different sites of metastases including liver ((**a**,**b**), black arrows), peritoneal carcinomatosis ((**b**), white arrows), retroperitoneal carcinomatosis (**c**), cutaneous metastases in front of pubis (**d**). In case of large retroperitoneal lymph nodes ((**e**), white arrows), a tumor thrombus in the inferior vena cava can be found ((**e**), black arrows). Spinal bone CT shows several vertebral metastases (**f**).

**Figure 3 cancers-14-03965-f003:**
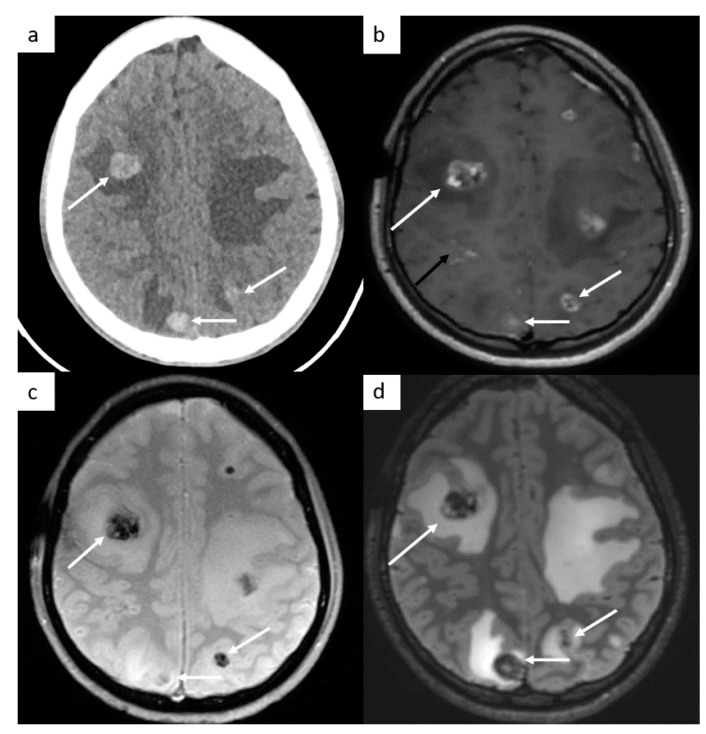
Brain metastases. Cerebral CT brain demonstrates several hemorrhagic metastases (**a**). MRI confirms the presence of several metastases with peripheral enhancement ((**b**), white arrows). They are hypointense on SWI-weighted images (**c**), with surrounding edema (**d**) on FLAIR-weighted images. In this case, a leptomeningitis is associated ((**b**), black arrow).

**Figure 4 cancers-14-03965-f004:**
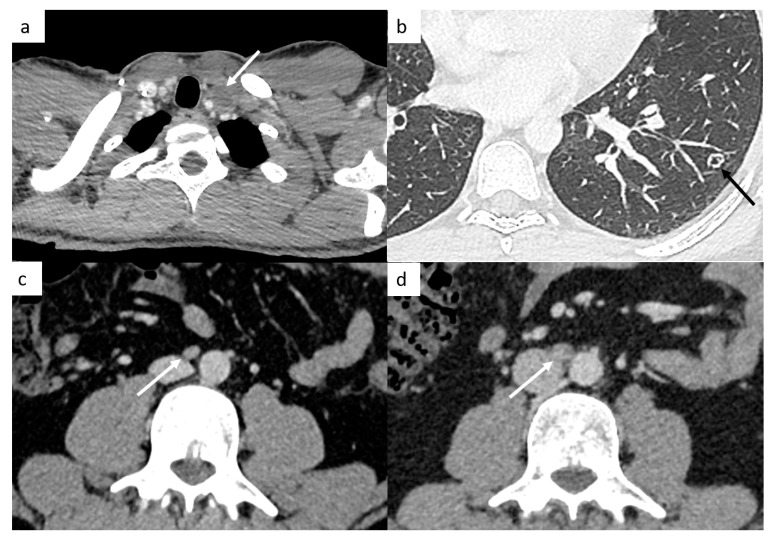
Small disease. Thoracic CT reveals a small left supraclavicular lymph node ((**a**), arrow). Lung metastases can appear as different features such as small nodules which may be excavated (**b**). Abdominal CT demonstrates a small interaorticocaval lymph node (**c**) and the repetition of the CT 6 months later shows an increasing size of this lymph node, a finding indicative of involvement (**d**).

**Figure 5 cancers-14-03965-f005:**
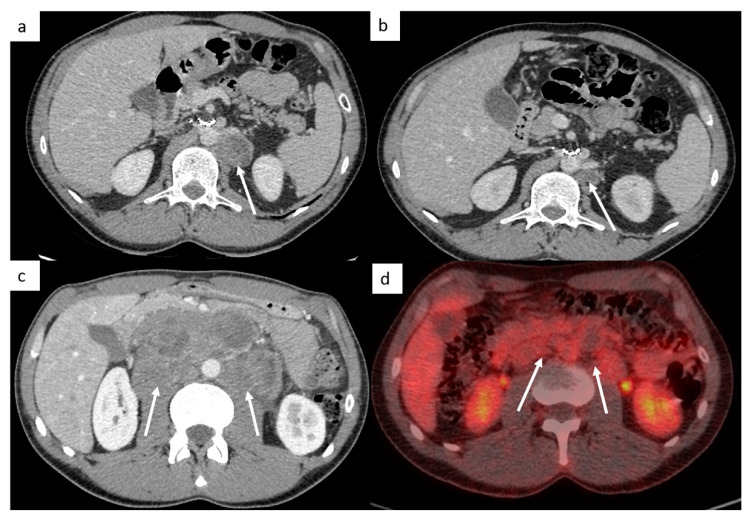
Post-chemotherapy lymph node changes. Abdominal CT reveals a left paraaortic lymph node measuring 27 mm (short axis) at initial staging ((**a**), arrow). After chemotherapy, CT shows a decrease in the size of this lymph node measuring 14 mm (short axis) ((**b**), arrow). Lymphadenectomy was performed, and histological examination of the surgical retroperitoneal resection specimen revealed teratomatous tissue and inflammatory changes, without any residual nodal invasion. Imaging in patient with left testicle seminoma (**c**,**d**). Abdominal CT demonstrates extensive retroperitoneal lymph nodes encasing the aorta at initial staging (**c**); follow-up evaluation after chemotherapy shows significant treatment response: lymph nodes decrease in size (<3 cm) and the FDG-PET CT confirms an excellent response, without increased activity in residual mass (**d**).

**Figure 6 cancers-14-03965-f006:**
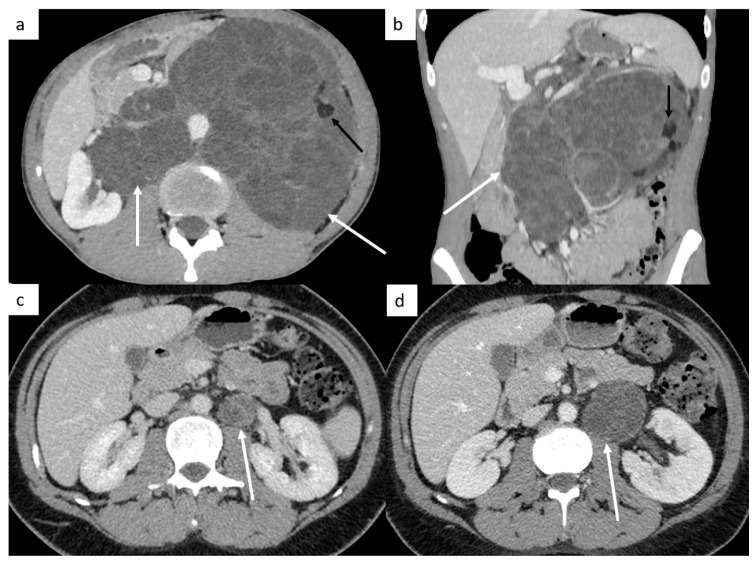
Teratoma and growing teratoma. Imaging in patient with a mature teratoma of the left testicle. Abdominal CT reveals a retroperitoneal mass ((**a**,**b**) white arrows) with fat tissue ((**a**,**b**) black arrows) corresponding to a teratoma. Abdominal CT shows a left paraaortic lymph node (**c**) with an increasing size during chemotherapy with classical features of growing teratoma: better circumscribed margins, expanding cystic and necrotic appearance of the lesion (**d**).

**Figure 7 cancers-14-03965-f007:**
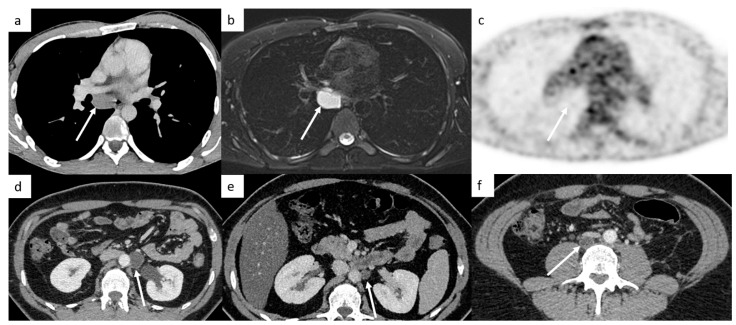
False-positive diagnoses. Thoracic CT shows a low-density mass below the carina in the middle mediastinum (**a**). A metastatic lymph node was suspected; however, this patient did not have any retroperitoneal lymph nodes. An MRI shows a well-circumscribed mass below the carina that appears hyperintense on T1-weighted images, hyperintense on T2-weighted images (**b**), without hypersignal on diffusion-weighted images. The FDG-PET CT confirms the absence of increased activity in this mass (**c**). It is a bronchogenic cyst. Post-surgical abdominal CT demonstrates a left paraaortic lymphocele (**d**) decreasing in size 3 months later, (**e**) but two years later, a low-density retrocaval mass appeared corresponding to teratoma (**f**). It shows that it can be difficult to differentiate teratoma from lymphocele, especially without prior CT to assess evolutions.

**Figure 8 cancers-14-03965-f008:**
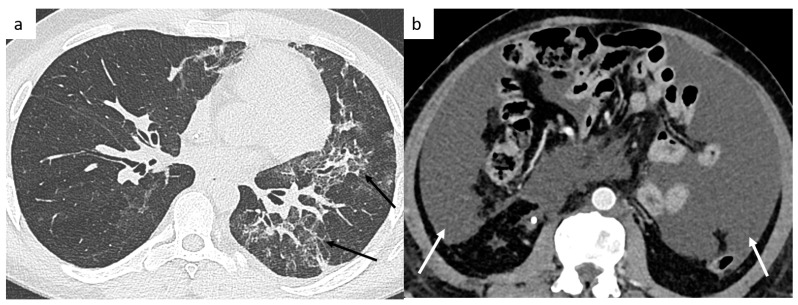
Post-treatment complications. Thoracic CT shows a bleomycin-induced interstitial pneumonitis (**a**) affecting left upper and lower lobes. Postoperative abdominal CT shows chylous ascites (**b**).

**Table 1 cancers-14-03965-t001:** TNMS staging for testicular cancer [16].

**pT**	**Primary Tumor**		
pTx	Primary tumor cannot be assessed		
pT0	No evidence of primary tumor (e.g., histological scar in testis)		
pTis	Intratubular germ cell neoplasia (carcinoma in situ)		
pT1	Tumor limited to testis and epididymis without vascular/lymphatic invasion; tumor may invade tunica albuginea but not tunica vaginalis		
pT2	Tumor limited to testis and epididymis with vascular/lymphatic invasion, or tumor extending through tunica albuginea with involvement of tunica vaginalis		
pT3	Tumor invades spermatic cord with or without vascular/lymphatic invasion		
pT4	Tumor invades scrotum with or without vascular/lymphatic invasion		
**N**	**Regional lymph nodes**		
Nx	Regional lymph nodes cannot be assessed		
N0	No regional lymph node metastasis		
N1	Metastasis with a lymph node mass 2 cm or less in greatest dimension or 5 or fewer positive nodes, none more than 2 cm in greatest dimension		
N2	Metastasis with a lymph node mass more than 2 cm but not more than 5 cm in greatest dimension; or more than 5 nodes positive, none more than 5 cm; or evidence of extranodal extension of tumor		
N3	Metastasis with a lymph node mass more than 5 cm in greatest dimension		
**M**	**Distant Metasasis**		
Mx	Distant metastasis cannot be assessed		
M0	No distant metastasis		
M1	Distant metastasis M1a Non-regional lymph node(s) or lung metastasis M1b Distant metastasis other than non-regional lymph nodes and lung		
**S**	**Serum tumor markers**		
Sx	Serum marker studies not available or not performed		
S0	Serum marker study levels within normal limits		
	**LDH (U/L)**	**HCG (mIU/mL)**	**AFP (ng/mL)**
S1	<1.5 × N and	<5000 and	<1000
S2	1.5–10 × N or	5000–50,000 or	1000–10,000
S3	>10 × N or	>50,000 or	>10,000

**Table 2 cancers-14-03965-t002:** Prognostic groups for testicular cancer [13].

Stage Grouping	T	N	M	S
Stage 0	pTis	N0	M0	S0
Stage I	pT1–T4	N0	M0	SX
Stage IA	pT1	N0	M0	S0
Stage IB	pT2–pT4	N0	M0	S0
Stage IS	Any pT/TX	N0	M0	S1–3
Stage II	Any pT/TX	N1–N3	M0	SX
Stage IIA	Any pT/TX	N1	M0	S0
	Any pT/TX	N1	M0	S1
Stage IIB	Any pT/TX	N2	M0	S0
	Any pT/TX	N2	M0	S1
Stage IIC	Any pT/TX	N3	M0	S0
	Any pT/TX	N3	M0	S1
Stage III	Any pT/TX	Any N	M1a	SX
Stage IIIA	Any pT/TX	Any N	M1a	S0
	Any pT/TX	Any N	M1a	S1
Stage IIIB	Any pT/TX	N1–N3	M0	S2
	Any pT/TX	Any N	M1a	S2
Stage IIIC	Any pT/TX	N1–N3	M0	S3
	Any pT/TX	Any N	M1a	S3
	Any pT/TX	Any N	M1b	Any S

**Table 3 cancers-14-03965-t003:** Updated prognostic-based staging system for metastatic germ cell cancer (IGCCCG) [18,19,20].

**Good-prognosis group**	
Non-seminoma 5-year PFS 92% 5-year survival 96%	All of the following criteria: Testis/retro-peritoneal primary No non-pulmonary visceral metastases AFP < 1000 ng/mL hCG < 5000 IU/L (1000 ng/mL) LDH < 1.5 × ULN (upper limit of normal)
Seminoma with LDH < 2.5 ULN 3-year PFS 92% and 93% in training and validation set 3-year survival 97% and 99% in training and validation set	All of the following criteria: Any primary site No non-pulmonary visceral metastases Normal AFP Any hCG LDH within 2.5 × ULN
Seminoma with LDH > 2.5 ULN 3-year PFS 80% and 75% in training and validation set 3-year survival 92% and 96% in training and validation set	All of the following criteria Any primary site No non-pulmonary visceral metastases Normal AFP Any hCG LDH > 2.5 × ULN
**Intermediate-prognosis group**	
Non-seminoma 5-year PFS 78% 5-year survival 89%	Any of the following criteria Testis/retro-peritoneal primary No non-pulmonary visceral metastases AFP 1000–10,000 ng/mL or hCG 5000–50,000 IU/L or LDH 1.5–10 × ULN
Seminoma 3-year PFS 78% and 61% in training and validation set 3-year survival 93% and 80% in training and validation set	All of the following criteria Any primary site Non-pulmonary visceral metastases Normal AFP Any hCG Any LDH
**Poor-prognosis group**	
Non-seminoma 5-year PFS 54% 5-year survival 67%	Any of the following criteria Mediastinal primary Non-pulmonary visceral metastases AFP > 10,000 ng/mL or hCG > 50,000 IU/L or LDH > 10 × ULN

## Data Availability

Not applicable.

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
