# Peer review of "The Role of CT in the Staging and Follow-Up of Testicular Tumors: Baseline, Recurrence and Pitfalls"

_cancers, 2022, doi:10.3390/cancers14163965_

Round 1
Reviewer 1 Report
In their review article, authors aimed to investigate the value of CT in testicular tumors for staging and follow-up. They have performed a comprehensive review on the baseline assessment, recurrence and pitfalls in staging and follow-up for testicular tumors. The referenced literature is adequate, and I have read the review with great interest. Overall, this is a thorough review. However, some minor details need to be addressed to strengthen the overall manuscript: Tables in the introduction are unclear, spacing needs to be addressed, i.e. it is not clear to the unexperienced reader what stage refers to which TNMS in table 2, especially for stage II and IIA.
Reviewer 2 Report
The review entitled “CT and testicular tumors: staging and follow-up: baseline, recurrence and pitfalls” is a well-written, systematically organized manuscript.
Authors should cite and discuss European guidelines for computed tomography-based testicular cancer staging and monitoring.
https://uroweb.org/guidelines/testicular-cancer
Reviewer 3 Report
I do not have any comments to made, the review is completed and easy to read.
Author Response
Dear reviewer,
Thank you for your nice comments.
Reviewer 4 Report
General comment
The manuscript entitled “CT and testicular tumors: staging and follow-up: baseline, recurrence and pitfalls” aims to review and summarize the role of CT in primary testicular germ cell tumors, focusing on its role during different stages of the disease. The manuscript is well written and the topic is interesting. It recalls a chapter of an academical book in terms of writing and linearity and authors should be praised for their work. Few corrections and suggestions are reported below, in order to improve furtherly, the manuscript. In addition, revise the paper to correct different typos along the way.
- Major issues
TITLE
The title is on point but the punctuation should be revised. Avoid the two colons
INTRODUCTION
Regarding epidemiology and risk factor of testicular cancer I suggest you to see: DOI: 10.3390/ijerph18168500 and DOI: 10.1210/clinem/dgab523
In addition, hypospadias and decreased fertility are not seen as proper risk factor for testicular cancer and should be deleted.
Considering the role of surgical treatment in testicular cancer, I suggest you to briefly report the therapeutic choices based on the type of testicular cancer. You could also see urological guidelines (see EAU guidelines) for improving this section.
FOLLOW UP AND RECURRENCE
In my opinion, the issue of repeated CT in young patients should be better discussed. Despite the beneficial effects of MRI on the reduction of ionizing radiations exposure, this imaging methodology has some limitations compared to CT. Please extend this discussion.
- Minor issues
BASELINE AND STAGING
2.1: Add references regarding the spread to lymphnodes of the disease.
BASELINE AND STAGING
Before introducing the role of CT, clarify how the suspected and certain diagnosis is made for testicular cancer.
CONCLUSION
Future perspectives should be reported.
Round 2
Reviewer 4 Report
The authors improved the manuscript accordingly. No further corrections are required in my opinion